# Management of acute kidney disease as part of routine clinical care in low-resource settings: The International Society of Nephrology Kidney Care Network Project

Rhys D.R. Evans[1],*, Sanjib K. Sharma[2], Rolando Claure-Del Granado[3,4], Brett Cullis[5], Emmanuel A. Burdmann[6], Fos Franca[6], Junio Aguiar[7], Martyn Fredlund[8], Kelly Hendricks[9], Maria F. Iturricha-Caceres[10], Mamit Rai[2], Bhupendra Shah[2], Shyam Kafle[2], David C. Harris[11], Mike V. Rocco[12]

**1** Centre for Kidney and Bladder Health, University College London, Royal Free Hospital, London, United Kingdom, **2** B.P. Koirala Institute of Health Sciences, Dharan, Nepal, **3** IIBISMED, Universidad Mayor de San Simon, School of Medicine, Cochabamba, Bolivia, **4** Division of Nephrology, Hospital Obrero No 2 – CNS, Cochabamba, Bolivia, **5** University of Cape Town, Cape Town, South Africa, **6** LIM 12, Division of Nephrology, and Department of Infectious and Parasitic Diseases, University of Sao Paulo, Medical School, Sao Paulo, Brazil, **7** University of Para, Santarem, Brazil, **8** North Bristol NHS Trust, Bristol, United Kingdom, **9** ISN Programs, Denver, Colorado, United States of America, **10** Facultad de Medicina, Universidad Privada del Valle, Tiquipaya, Bolivia, **11** Westmead Institute for Medical Research, University of Sydney, Sydney, Australia, **12** Wake Forest School of Medicine, Winston-Salem, North Carolina United States of America

* Rhys.evans5@nhs.net

## Abstract

Acute Kidney Disease (AKD) commonly affects disadvantaged populations in low-resourced areas with poor access to kidney care. Here, barriers to management include a lack of AKD education alongside an inability to measure serum creatinine (SCr) to identify kidney disease. The Kidney Care Network (KCN) is a service improvement initiative which aims to implement a novel strategy for the management of AKD into routine clinical care in low- and low-middle income countries (LLMICs). The strategy includes the development of a scoring system to screen patients for risk of AKD and the use of a device to measure SCr at the point-of-care (POC). This approach is underpinned by dedicated AKD training activities for healthcare workers providing front line clinical care. We report feasibility in the implementation of the KCN approach in adults in 4 LLMICs. Between 2018–2020, 4311 patients at project sites in Bolivia, Brazil, Nepal, and South Africa were deemed at risk of kidney disease and underwent SCr testing, predominantly with the POC device. AKD was identified in 2922 (67.8%) patients. AKD was most commonly due to infections and hypo-volemia, and as such was treatable by relatively simple means. Most patients with AKD were treated at the site of patient presentation, including rural primary healthcare facilities, and with early AKD identification the need for kidney replacement therapy was low. In-hospital mortality was only 2.9% and follow-up occurred at 3 months in 1865 (62.3%) patients discharged post AKD diagnosis. Hence, we show the KCN approach is a feasible and effective mechanism for improving AKD management in LLMICs.

**Data availability statement:** The data underlying the results presented in the study are owned by the International Society of Nephrology and will be made freely available on reasonable request (research@theisn.org).

**Funding:** The Stavros Niarchos Foundation funded the study; Nova Biomedical provided in-kind support. The funders were not involved in any aspect of the study beyond this, including design, data collection and analysis. No payment has been provided to write this article.

**Competing interests:** I have read the journal's policy and the authors of this manuscript have the following competing interests: RE has received honoraria from Therakos. EB has received honoraria from Baxter and AstraZeneca.

## Introduction

Episodes of acute reduction in excretory kidney function, termed Acute Kidney Disease (AKD), commonly occur in diverse populations worldwide. AKD is characterized by a reduction in glomerular filtration rate (GFR), which has been present for less than 3 months. A subset of patients with AKD will fulfil the diagnostic criteria for Acute Kidney Injury (AKI), and 13.3 million cases of AKI are estimated to occur globally each year, leading to 1.7 million deaths [1,2]. Whilst a lack of registry data at the population level in Low- and Low-middle income countries (LLMICs) make the precise prevalence of AKD difficult to define, the majority of AKD cases and subsequent deaths are thought to occur in disadvantaged communities in low-resourced settings with poor access to care [2,3]. In response to a concern of avoidable young deaths as a result of AKD, the International Society of Nephrology (ISN) launched the 0by25 initiative in 2013, which advocates that zero people should die of untreated AKD in the poorest parts of Africa, Asia and Latin America by 2025 [2,4]. The initiative proposes that management of AKD in low-resource regions should become a human right, akin to the administration of antiretroviral drugs to treat people living with HIV [5]. The overarching aims of 0by25 are to establish the global burden of AKD, raise awareness and reduce variation in AKD care, and create a sustainable infrastructure for its management worldwide.

A series of project cycles has been undertaken to achieve these aims [6]. To better understand the epidemiology of AKD in LLMICs, a multinational cross-sectional study termed the 'Global Snapshot' was undertaken [7–9]. This highlighted that the majority of AKD in LLMICs was associated with treatable conditions such as dehydration, hypovolemia and infection. It confirmed previous concerns of higher AKD mortality in low- compared to high-resource settings and reinforced the reduced availability of kidney replacement therapy (KRT) in LLMICs. Based on these results, a protocol to identify and manage AKD was developed [10]. This process included the development of a symptom-based scoring system to assess risk of AKD, and the use of point-of-care (POC) serum creatinine (SCr) testing to screen for kidney disease. This approach was underpinned by education and training of healthcare workers in the management of AKD, and its feasibility was proven in a pilot study in 3 LLMICs [10].

The Kidney Care Network (KCN) project is the most recent initiative within the 0by25 framework. Its aim is to implement a strategy based on education and POC SCr testing to identify and manage AKD as part of routine clinical care in LLMICs. Herein, we describe the feasibility of project implementation in adults in 4 LLMICs. In this narrative summary, we report both successes and challenges with the approach, and provide recommendations for the future use of the strategy in AKD care.

## Project description

### Overview of the KCN approach

The KCN project was implemented in low-resourced regions of Brazil, Bolivia, South Africa and Nepal between 1st September 2018 and 30th November 2020. The project was undertaken in adults and a variety of healthcare facility types, including healthcare centers (HCCs), district hospitals, and tertiary hospitals, within each country were selected as sites for project implementation (S1 Table). HCCs act as primary healthcare facilities in these regions and provide only basic care. Staffing and supplies are limited, and access to blood tests (specifically SCr) is not routinely available. HCCs are often in rural areas including in this project the Amazon region of Brazil and remote sites in northern KwaZulu-Natal, South Africa. District and tertiary hospitals provide increasingly specialized care including the provision of KRT at the tertiary level. However, more sophisticated diagnostics including kidney biopsy were

largely unavailable and there was only limited access to critical care for multi-organ support as would be routine in higher-income settings.

Fig 1 provides a graphical summary of the KCN approach. In each country, implementation of the approach was managed by project leads (SS, BC, EB, RDG), who were supported by the KCN leadership team (RE, KH, DH, MR) within the ISN. In the first stage of the project, healthcare workers providing clinical care at each site underwent an education and training program on the identification and management of AKD. The training was site specific and most often delivered in face-face workshops over multiple days. Subsequently, a protocol to identify AKD was instituted at each site, which included the use of a symptom-based risk score to aid AKD identification. The risk score was developed using a regression analysis to determine the clinical variables associated with AKD in the previous 0by25 Pilot Feasibility Study [10]. Points within the scoring system are attributed to symptoms associated with AKD (Table 1). Patients presenting with a risk score of 10 points or more were considered to be at risk of AKD and underwent SCr testing. Patients with a risk score of <10 points could also be considered at risk of AKD and undergo SCr testing according to the judgment of the clinical team.

After screening, measurement of SCr was predominantly undertaken at the point-of-care using devices that were provided to each site as part of the project (StatSensor Xpress CREA, Nova Biomedical, Waltham, MA, US) [11,12]. These are small hand-held devices, which are powered by a replaceable 3V lithium button battery. They use test strips that need to be stored at 4–8 degrees Celsius and have a shelf life of 12 months.

SCr was used to estimate GFR (eGFR) using the CKD-EPI equation without race adjustment [13,14]. Both results were used to identify patients with kidney disease according to Kidney Disease Improving Global Outcomes (KDIGO) functional criteria and classified as either having acute or chronic kidney disease (AKD or CKD) (S2 Table). In accordance with the latest KDIGO consensus statement, AKD was defined by 'abnormalities of kidney

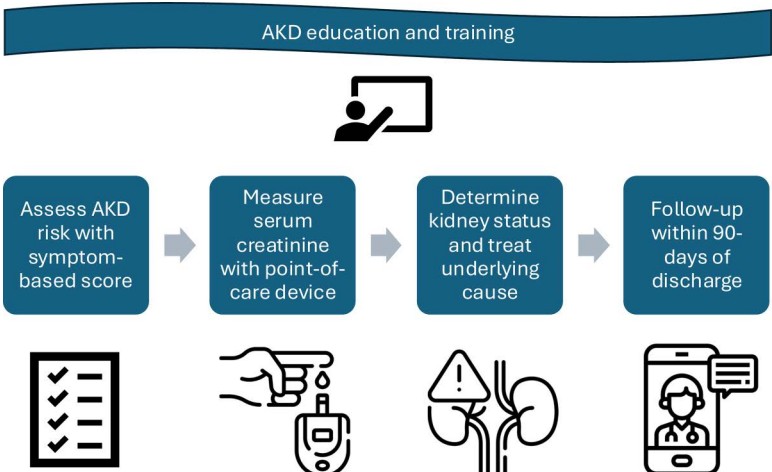

**Fig 1. Graphical summary of the Kidney Care Network Approach.** The project was underpinned by education and training of healthcare workers in the management of AKD. Patients presenting to healthcare facilities were screened for risk of AKD using a symptom-based risk score, alongside clinical judgement. Those at increased risk underwent serum creatinine testing with a point-of-care device. Kidney status was then determined, and the potential causes of AKD identified and treated. Management was predominantly within the healthcare facility of patient presentation, but in severe AKD cases, patients were transferred to a higher-level facility for more specialist care. Patients were followed-up to reassess kidney status within 3 months of discharge.

**Table 1. AKD risk score components. The area under the receiver operating characteristic curve was 0.824 for the risk score to detect AKD with an optimal cut-off score of 10 points (sensitivity 92.9% and specificity 58.9% at this cut-off) based on data from the 0by25 Pilot Feasibility cohort. As such, a score of 10 points or more is considered to represent increased risk of acute kidney disease.**

| Factor | Points |
|---|---|
| Vomiting | 4 |
| Low oral Intake | 2 |
| Weakness | 2 |
| Oliguria reported by patient | 8 |
| Hypotension | 8 |
| Appetite Loss | 8 |
| Swelling | 5 |

Variable description:

Vomiting – presence of dehydration associated with vomiting as determined by clinical team.

Low oral intake – presence of dehydration associated with low oral intake as determined by clinical team.

Weakness – reported by patient.

Oliguria – reported by patient.

Hypotension – blood pressure <90/60 mmHg or relative hypotension as determined by clinical team.

Loss of Appetite – acute or chronic symptom reported by patient.

Swelling– presence of non-traumatic swelling on limbs, face or entire body.

function and/or structure with a duration of < 3 months'; it was separated into AKD with and without AKI [15]. AKI was diagnosed and staged according to KDIGO criteria [16]. The latest SCr documented prior to healthcare facility admission and the lowest SCr during healthcare facility admission were used to determine the baseline SCr; an imputed baseline SCr based on an assumed eGFR was not used [17]. Urine output measurement and urinalysis data were not captured. Patients were categorized into those with and without kidney disease, and the nature of kidney disease was determined: AKD with AKI; AKD without AKI; or CKD.

Patients with suspected kidney disease were managed according to the discretion of the treating clinician who had previously undergone training in the initial stages of the project. Serial SCr measurements were made in some patients and follow-up of patients with kidney disease was encouraged within 90 days of hospital discharge. The project included both patients who attended a healthcare facility, underwent creatinine testing (+- the relevant management) and were discharged on the same day, in addition to those patients that were admitted and then discharged after an inpatient stay.

## Ethics statement

Ethics approval was granted locally at each of the four study sites by the following ethics boards: the ethical committee of the Escola de Enfermagem da USP (University of Sao Paulo Nursing School), Brazil, approval 31670214; The Comité Regional de Enseñanza e Investigación, Hospital Obrero No 2 - Caja Nacional de Salud, Cochabamba, Bolivia; the Nepal Health Research Council, Kathmandu, approval 205/2016; and the UKZN biomedical research ethics committee, South Africa, approval BE257/19. Consent was written in Brazil and Nepal, and verbal in Bolivia. The requirement for consent was waived by the ethics board in South Africa as the project was categorized as a service improvement initiative.

## Summary of the clinical features of patients managed within the KCN Project

A minimum clinical dataset was recorded to permit assessment of project implementation but not to detract significantly from the overarching aim of embedding the KCN approach within routine clinical care. 4394 patients aged ≥ 18 years were successfully screened across the project sites and 4311 patients were deemed at risk of kidney disease and managed according the KCN protocol (Fig 2). The largest number of patients was managed in Nepal (n = 1952) and the least in Brazil (n = 197) (Table 2). 2289 (53.1%) patients were female, and median age was 57 (42–70) years. SCr was measured by POC device in 3145 (73.0%) patients; the remaining patients had SCr measured in the local laboratory by automated analyser. Laboratory measurement of creatinine was most common in Bolivia. Kidney disease was present in 2959 (68.6%) patients, which included 2922 (67.8%) patients with AKD and 37 (0.9%) patients with CKD.

Concordant with findings from previous reports, the most common causes of AKD were infection and hypovolemia; management consisted most often of antimicrobials and fluid

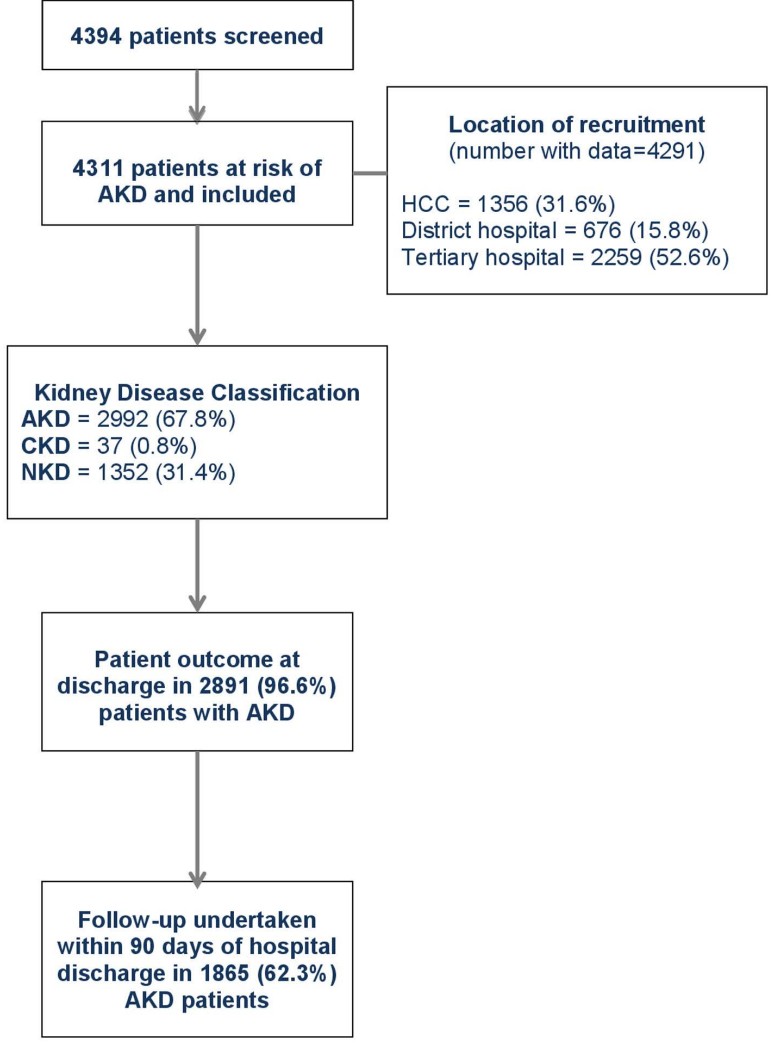

**Fig 2. KCN project cohort description.**

**Table 2. Clinical features of patients managed as part of the KCN project.**

| | BOLIVIA | BRAZIL | NEPAL | SOUTH AFRICA | ALL PATIENTS |
|---|---|---|---|---|---|
| **Screening** | | | | | |
| Screened (n) | 1010 | 199 | 1963 | 1222 | **4394** |
| Excluded (n; % of screened) | 59 (5.8) | 2 (1.0) | 11 (0.6) | 11 (0.9) | **83 (1.9)** |
| Included (n; % of screened) | 951 (94.2) | 197 (99.0) | 1952 (99.4) | 1211 (99.1) | **4311 (98.1)** |
| Included with risk score ≥10 points (n; % of included) | 951 (100.0) | 160 (81.2) | 1947 (99.7) | 132 (10.9) | **3190 (74.0)** |
| Included on basis clinical judgment (risk score < 10 points) (n; % of included) | 0 (0.0) | 37 (18.8) | 5 (0.3) | 1079 (89.1) | **1121 (26.0)** |
| Risk score (median; IQR) | 16 (13-20) | 15 (10-20) | 18 (14-22) | 2 (0-8) | **14 (8-19)** |
| Healthcare facility type where patient enrolled | | | | | |
| Healthcare center | 113 (11.9) | 50 (25.4) | 0 (0.0) | 1193 (100.0) | **1356 (31.6)** |
| District Hospital | 531 (55.8) | 145 (73.6) | 0 (0.0) | 0 (0.0) | **676 (15.7)** |
| Tertiary Hospital | 307 (32.3) | 0 (0.0) | 1952 (100.0) | 0 (0.0) | **2259 (52.6)** |
| Other/missing | 0 (0.0) | 2 (1.0) | 0 (0.0) | 0 (0.0) | **2 (0.0)** |
| **Demographics** | | | | | |
| Female (n; %) | 517 (54.4) | 88 (44.7) | 880 (45.1) | 804 (66.4) | **2289 (53.1)** |
| Age (median; IQR) | 56 (39-69) | 57 (42-69) | 60 (42-71) | 55 (43-66) | **57 (42-70)** |
| **Measurement of kidney function** | | | | | |
| SCr measured by POC device (n; %) | 431 (45.3) | 194 (98.5) | 1309 (67.2) | 1211 (100) | **3145 (73.0)** |
| Enrollment SCr (mg/dL) (median; IQR) | 1.4 (1.0-1.9) | 1.2 (0.9-1.8) | 1.7 (1.4-2.3) | 1.0 (0.8-1.2) | **1.4 (1.0-1.9)** |
| Enrollment eGFR (ml/min/1.73m²) (median; IQR) | 51 (33-75) | 60 (36-85) | 41 (28-56) | 75 (57-93) | **52 (34-78)** |
| **Classification of kidney disease** | | | | | |
| Number with data | 951 | 197 | 1952 | 1211 | **4311** |
| AKD with AKI (n; %) | 116 (12.2) | 14 (7.1) | 404 (20.7) | 100 (8.3) | **634 (14.7)** |
| AKI Stage 1 (n; % of AKI) | 60 (51.7) | 7 (50.0) | 235 (58.2) | 89 (89.0) | **391 (61.7)** |
| AKI Stage 2 (n; % of AKI) | 39 (33.6) | 1 (7.1) | 93 (23.0) | 7 (7.0) | **140 (22.1)** |
| AKI Stage 3 (n; % of AKI) | 17 (14.7) | 6 (42.9) | 76 (18.8) | 4 (4.0) | **103 (16.3)** |
| AKD without AKI (n; %) | 568 (59.7) | 123 (62.4) | 1295 (66.3) | 302 (24.9) | **2288 (53.1)** |
| CKD (n; %) | 28 (2.9) | 4 (2.0) | 5 (0.3) | 0 (0.0) | **37 (0.9)** |
| NKD (n; %) | 239 (25.1) | 56 (28.4) | 248 (12.7) | 809 (66.8) | **1352 (31.4)** |

AKD – Acute Kidney Disease; AKI – Acute Kidney Injury; CKD – Chronic Kidney Disease; NKD – No Kidney Disease; SCr – Serum creatinine; eGFR – estimated glomerular filtration rate.

replacement. Most patients were managed in the facility where they presented; transfer to a higher-level healthcare facility was recommended in only 178 patients and transfer occurred in 132 (74.2%) cases. KRT was indicated in 32 (1.1%) patients and provided in 26 (81.3%) patients. Hemodialysis was the KRT modality used in all cases. Reasons for not providing KRT included patient inability to afford treatment (n = 1) and refusal of treatment for non-financial reasons (n = 5). Eighty-four (2.9%) patients with AKD died during their hospital admission and 1865 (62.3%) patients were followed-up by the management team within 90 days of discharge.

## Country level case studies of project implementation

In the sections that follow, we outline project implementation in more detail in each of the 4 participating countries. We provide more granularity on each stage of the KCN approach at the individual sites. Key findings are summarised in **Table 3**.

 **Bolivia.** The project was implemented in a mix of healthcare facility types (3 HCCs, 3 district hospitals, and 1 tertiary hospital) in both rural and urban areas of Cochabamba,

Table 3. Summary of KCN project implementation.

| | BOLIVIA | BRAZIL | NEPAL | SOUTH AFRICA |
|---|---|---|---|---|
| **SETTING FOR PROJECT IMPLEMENTATION** | | | | |
| **Healthcare facilities** | Mix of HCCs (n = 3), district hospitals (n = 3) and a tertiary hospital (n = 1), in rural and urban areas of central Bolivia | One district hospital (urban) and 3 HCCs (2 rural, 1 urban) in the Amazon region of north Brazil. | Tertiary hospital only (n = 1), in urban area of south-east Nepal. | Predominantly HCCs (n = 7); also, one district hospital in rural northeast South Africa |
| **Key population characteristics** | N = 1010 Approx. 80% managed with public funding, 20% with insurance | N = 199 100% managed with public funding. Some in remote areas. | N = 1963 100% managed with public funding | N = 1222 100% managed with public funding. Young, almost entirely black African cohort, with high prevalence HIV and high unemployment rate. |
| **Resources available for kidney care** | HCCs: no ability to measure serum creatinine prior to project Hospitals: tertiary and district hospitals with central laboratory and imaging with ultrasound available; all forms of kidney replacement therapy available at tertiary centre. | HCCs: no ability to measure serum creatinine prior to project. Staffed by nursing staff and assistants. District hospitals: central laboratory and imaging with ultrasound available; kidney replacement therapy available at district hospital | Central laboratory, imaging with ultrasound and kidney replacement therapy available | HCCs: no in-house laboratory (2-day turnaround for creatinine prior to project). Staffed by nurses and nursing assistants predominantly. Telemedicine support available from district hospital. District Hospital: laboratory and imaging available; acute peritoneal dialysis available for AKI. |
| **EDUCATION AND TRAINING** | | | | |
| **Events undertaken (number and format)** | Face-face workshops undertaken at the start of the project, then twice yearly thereafter. Delivered by Nephrologist and coordinators part of a kidney research team | 3 events in total, all face-face workshops with interactive discussions. One 2-day event within the district hospital, then two 1-day events in rural HCCs. | 1-day face-face workshop at the start of the project, repeated twice yearly. Delivered by nephrologist and general physicians | Training events (face-face workshops) at each of the project sites at the start, repeated after 2 months. Delivered by nephrology research registrar. Ongoing training thereafter by project nurse weekly. |
| **Attendees** | Medical students, residents, fellows, attending physicians, and nurses | Approx. 30 attendees at the hospital event including assistants, nurses, residents, fellows, and physicians. HCC teaching attended by local staff | Non-specialist doctors, nurses, and health assistants | All nurses and nursing assistants working within the HCCs. |
| **Comments/Challenges** | Teaching delivered at each site including in rural areas; positive feedback received | Positive feedback from each session | Positive feedback and desire for further training | Positive feedback. Only 20% of attendees had received AKD training previously. |
| **AKD SCREENING** | | | | |
| **Enrollment based on risk score** | Enrolment based on risk score in all patients; ongoing use after period of data collection reported | Enrolment based on risk score in 81% patients | Enrolment based on risk score in almost all (99.7%) patients | Enrolment based on risk score only in a minority of patients (11%); most patients included according to clinical judgement |
| **Creatinine measurement by POC device (n; %)** | 431 (45.3) | 194 (98.5) | 1309 (67.2) | 1211 (100) |
| **Number (%) of patients with AKD** | 684 (71.9) | 137 (69.5) | 1699 (87.0) | 402 (33.2) |
| **Comments/Challenges** | Screening easy to implement but additional workload | Some insecurity from the allied health team staffing the HCCs to enroll patients without doctor supervision | Screening easy to implement but some challenges in screening all patients in a busy emergency department. Additional workload. | Liberal inclusion of patients at HCCs in whom clinical team had suspicion of AKD irrespective of risk score. Very busy clinics with additional workload for screening. Usually, no medical staff on site for queries. Some challenges with cold storage of reagents. |

*(Continued)*

**Table 3.** (Continued)

| | BOLIVIA | BRAZIL | NEPAL | SOUTH AFRICA |
|---|---|---|---|---|
| **MANAGEMENT OF AKD** | | | | |
| **Main causes of AKD** | Infection 45%; dehydration/ hypovolemia 26%; Medication/ toxin 8% | Infection (35%); dehydration/ hypovolemia 14%; animal bite 13%; cardiac failure 13%; | Infection 44%; dehydration/ hypovolemia 13%; hepatic failure 10% | Dehydration/ hypovolemia 12%; infection 10%; medication/ toxin 10%; HIV 5% |
| **Treatment of AKD** | Intravenous fluid 72%; oral fluid 47%; antibiotics 60% | Antibiotics 46%; intravenous fluid 27%; diuretics 17%; anti-venom 8% | Antibiotics 86%; intravenous fluid 86%; diuretics 11% | Antibiotics 10%; intravenous fluid 8%; oral fluid 7%; HIV therapy 5% |
| **Transfer to a higher-level healthcare facility for treatment (n; % included patients)** | 46 (4.8) | 2 (1.0) | 0 (0.0) | 82 (6.8) |
| **Dialysis indicated (n; % of AKD)** | 24 (3.5) | 7 (5.1) | 1 (0.05) | 0 (0) |
| **Dialysis provided (n; % of indicated)** | 19 (79.2) | 6 (85.7) | 1 (100) | 0 (N/A) |
| **Comments/Challenges** | Causes identified and treated according to the STOP AKI schema; telehealth implemented to improve communication between levels of healthcare and facilitate transfer when needed | Snake bite as a cause specific to this region with lack of anti-venom availability in some cases; challenging travel from HCC to district hospital; lack of healthcare staff in HCCs, and high turnover rate when present; telehealth support for HCCs inconsistent | Lower requirement of dialysis than expected; and dialysis had to be paid for when required. | Lower levels of AKD than anticipated; and lower severity than previously encountered. |
| **AKD OUTCOMES AND FOLLOW-UP** | | | | |
| **AKD patient outcome at discharge** | 90% survival | 97% survival | 99% survival | 98% survival |
| **AKD kidney outcome at discharge** | 13% unknown; 41% incomplete/ no recovery | 17% unknown; 31% incomplete/no recovery | 56% unknown; 14% incomplete/no recovery | 72% unknown; 6% incomplete/no recovery |
| **Follow-up of AKD patients at 90 days (n; %)** | 490 (71.6) | 89 (65.0) | 1418 (83.5) | 122 (30.3) |
| **Patient outcome at 90-days in those followed-up** | 99% survival | 86% survival | 86% survival | 100% survival |
| **Kidney outcome at 90-days in those followed-up** | 37% with incomplete/no recovery | 68% with incomplete/no recovery | 52% with incomplete/no recovery | 49% with incomplete/no recovery |
| **Comments/Challenges** | Absence of electronic records making outcomes difficult to capture in routine care | Finance restrictions impacted patients attending for follow-up locally and specifically when required at district hospital (cost of travel) | Finance restrictions impacted follow-up (time off work). Higher rates of post-discharge mortality and persistent kidney disease than expected. | Finance restrictions and limitations in health literacy impacted follow-up. |

central Bolivia. Most of the population served by these facilities access healthcare through public funding. Prior to the project, HCCs had no ability to measure SCr; a central laboratory and ultrasound imaging capability were present in the hospitals, with all forms of KRT available at the tertiary centre. Many of the sites had participated in previous phases of 0by25, which facilitated implementation of the project; this was further helped by coordinators from

an established nephrology research and quality improvement team. Education and training were delivered in the form of face-face workshops at the start of the project and then twice yearly thereafter at each of the study sites, including those in rural areas, helping to ensure knowledge retention throughout the project. These sessions were attended by a cross-section of professionals from the multidisciplinary team, with positive feedback received. 1010 patients were screened for AKD risk, all with the use of the risk score, which was easy to implement but which led to some additional workload for the clinical teams. Approximately one half of SCr measurements were made with the POC device. A high proportion (72%) of patients screened had AKD, and aetiologies were identified using the acute kidney injury network 'STOP' (sepsis/hypoperfusion, toxins, obstruction, parenchymal kidney disease) schema, and treated accordingly [18]. Infection, hypoperfusion and toxins were the expected and most encountered causes. Management of AKD was predominantly undertaken at the site of patient presentation, with telehealth implemented to improve communication between levels of healthcare and facilitate patient transfer when needed. Established relationships between the project sites, and effective teamwork were important in this regard. Dialysis was indicated in only 3% of AKD patients and provided when needed in 79% of cases. In-hospital patient survival in those with AKD was 90%; 41% of AKD patients left hospital with either incomplete or no recovery of kidney function. 72% of AKD patients were followed up at 90-days post discharge and 37% had persistent kidney injury at this timepoint. In Bolivia, there is a publicly funded management program for patients with CKD, which includes coverage for KRT, and patients identified with a new diagnosis of CKD were managed within this framework.

**Brazil.** The project was implemented in the Amazon region of northern Brazil, in 3 HCCs, two of which were in rural areas, and a district hospital in Santarem. The entire population served in the region access healthcare using social security funding. The distance between the rural HCCs and the district hospital as well as the cost of the required travel between sites presented some challenges; travel from the HCC at Arapixuna to the district hospital requires a boat trip along the Amazon River. HCCs were most often run by nursing staff and health agents, and prior to the study these centres had no capability for SCr measurement or administration of intravenous fluids. The district hospital has an established laboratory and imaging available for the management of kidney disease. Three training events were undertaken: the first was a 2-day event in Santarem, which was attended by 30 members of the multidisciplinary team; the other two events were one-day workshops which were undertaken within the rural HCCs. There was enthusiasm across sites for the training. 199 patients were screened for risk of AKD, predominantly with the AKD risk score, and SCr was measured by POC device in 99% of cases. Despite training and the intermittent use of telehealth to provide support HCC staff, there was some initial insecurity from HCC staff in screening and managing patients with AKD, highlighting the need for additional support for healthcare workers in more remote areas. However, 70% of those who had creatinine measured were found to have AKD, demonstrating the effectiveness of the risk score specifically for use by those with limited experience of managing AKD. AKD was most commonly the result of infection and hypovolemia; envenomation was the aetiology in 13% of cases, and a cause specific to this region. 99% of AKD cases were managed within the site of patient presentation, most commonly with antibiotics and intravenous fluids; anti-venom was often but not universally available. Dialysis was indicated in only 5% of AKD cases and provided in most of these through public funding. Survival to hospital discharge occurred in 97% of patients; 31% of AKD patients left hospital with incomplete or no recovery of kidney function. 65% of AKD patients were followed up 90-days post discharge; financial restrictions in terms of the cost of travel back to the healthcare facility had some impact on follow-up rate. Survival to this

timepoint occurred in 86% of AKD patients albeit a particularly high proportion of surviving patients (68%) had persistent kidney disease.

**Nepal.** The project was implemented in a tertiary hospital in Dharan, a city in southeast Nepal. Here, there are facilities to measure SCr, undertake urinalysis, perform imaging, and provide KRT. All patients access healthcare through public funding. A 2-day face-face workshop was held at the start of the project, and then training events were repeated twice yearly thereafter. These were delivered by a nephrologist and two other physicians, and attended by non-specialist doctors, nurses, and health assistants. They were well received with requests for further training of a similar nature. 1963 patients were screened for AKD risk, 99.7% with the use of the risk score. Screening was easy to implement, albeit there was some apprehension in establishing its use and the additional workload led to some challenges, especially in a busy emergency department where many of the patients presented. Two thirds of patients had SCr measured by POC device and 1699 (87%) had AKD. Infection and hypoperfusion in the setting of hypovolemia and hepatic failure were the commonest causes of AKD; antibiotics, intravenous fluids and diuretics were the most frequent treatments. Only 1 patient required KRT, which was less than anticipated, and may reflect the early detection of kidney disease. In-hospital survival was 99% in patients with AKD; 56% of AKD patients were discharged without repeat measure of SCr. Despite concerns about loss of wages and the costs required for travel back to the hospital, a high proportion of patients (n = 1418, 83.5%) were followed-up post hospital discharge. There was a 14% mortality rate in the 3-month period post hospital discharge, which was higher than expected. 52% of patients had incomplete or no recovery of kidney function at this timepoint. Denial of CKD diagnosis was common at this stage and whilst some CKD care is available within the public health system, often this comes with significant delays, and patients commonly sought management within the private sector with associated out-of-pocket costs. This highlights the need to ensure capacity to provide kidney care across the continuum of AKD and CKD when implementing the KCN approach even though it is an initiative aimed primarily at improving AKD care.

**South Africa.** The project in South Africa was implemented predominantly in HCCs (n=7) in rural parts of KwaZulu-Natal. The district hospital at Mseleni provided support in the management of patients with AKD, but patient identification occurred entirely within the HCCs. The population served by these healthcare facilities is young, almost entirely black African, with high rates of unemployment (70.5–87%), and prevalent HIV (44%). Prior to project implementation, SCr measurement was not available within the HCCs themselves; a serum sample could be sent for analysis to a central laboratory but the turnaround time for the result was at least 48 hours once transport of the sample to the laboratory had been factored in. The district hospital has an onsite laboratory for SCr testing, in addition to imaging and capability to provide acute peritoneal dialysis. HCCs are run primarily by nursing staff and assistants, and a training event was undertaken at each clinic at the start of the study and repeated at 2 months thereafter. The same occurred at the district hospital, with ongoing training provided as needed throughout the project (up to weekly at times). This was in the form of 1-hour workshops delivered by a nephrology fellow and a dedicated project nurse who coordinated project implementation across the South African sites. Only 20% of attendees at the education events had prior training in AKD management, and the sessions were positively received. Risk of AKD was determined by the nursing teams within each HCC. Unlike other sites, clinical judgement was used as the predominant mechanism for determining AKD risk, with the risk score only being used in 11% of patients. Of the 1211 patients thought to be at risk, only 402 (33.2%) had AKD, significantly less than sites where the risk score was used, highlighting its effectiveness. Clinics were very busy, and the additional workload required for screening may have impacted the approach to determining AKD risk, in addition to the

lack of onsite clinicians to help. All patients had SCr measured by POC device, which was easy to undertake albeit there were occasional challenges with cold storage of reagents. Infection (including HIV), hypovolemia and toxins were the commonest causes of AKD, and most cases were managed within HCCs, highlighting that with the relevant training and resources for early detection and monitoring, AKD can be successfully managed in rural primary care settings. 7% of patients were transferred to the district hospital, but none of these patients required KRT. 98% of AKD patients survived to healthcare facility discharge; the majority (78%) of patients were discharged without repeat measure of creatinine and as such for most kidney recovery was unknown. There were specific challenges undertaking follow-up with only 30% of AKD patients reviewed post HCC discharge; financial restrictions and low rates of health literacy were likely contributors. In patients who underwent follow-up, 49% had persistent kidney disease, highlighting the need to develop specific strategies to facilitate follow-up in the most rural areas and avoid missed opportunities to identify and then prevent progression of CKD.

## Discussion

The KCN proposes a novel approach for the management of AKD for use in LLMICs. This is needed due to the high burden of kidney disease present in LLMICs, and a current lack of capacity for its identification and treatment, ultimately leading to avoidable patient deaths. We have described the feasibility of implementing the KCN approach in a large number of patients from diverse populations presenting to a variety of healthcare facility types in 4 countries across 3 continents. Over 50% of participants were female, and hence the approach may promote more equitable kidney care. Its implementation occurred in both urban and rural areas, including in facilities where measurement of SCr was not previously possible, staffed by healthcare workers with limited or no previous training in the management of AKD. Specifically, the approach was feasible and effective in HCCs, and its potential to transform kidney care may be greatest in patients presenting to these healthcare facility types. As outlined in our report, the main constituents of the approach are the use of a symptom-based score to identify patients at risk of AKD, the measurement of SCr with a POC device to confirm its presence, the identification and treatment of the underlying cause of AKD, with follow-up of kidney status thereafter in both the short and medium term. These are underpinned with education and training activities on AKD, undertaken both prior to and during the project.

As we have highlighted previously, education was most needed in rural settings where healthcare staff have the least training in the management of AKD [19]. It was most successful when repeated regularly throughout the project. Establishing good relationships between the staff in HCCs and those in higher level healthcare facilities was important in providing coordinated AKD care, in addition to effective telehealth to support primary healthcare providers who had limited previous experience in managing AKD. The presence of local clinical leaders that championed AKD care was key for project implementation in its entirety.

The use of the risk score followed by the measurement SCr by POC device were easy to implement, even in healthcare settings that were not used to managing AKD. This was highlighted by the successful continuation of the project during the early stages of the COVID-19 pandemic when more senior healthcare expertise was diverted elsewhere. The use of the risk score added some additional workload at the time of patient presentation, but this was offset by the early identification of AKD, and hence less time and resource were required to manage complications of more advanced AKD thereafter. A full analysis of its effectiveness is beyond the scope of this report; however, it was notable that AKD was much less frequent in patients included according to clinical judgement as opposed to when included with the use of the risk score.

The point-of-care device enabled the measurement of SCr within minutes, leading to immediate assessment of kidney function. As a result, patients with AKD could be identified in the early stages, facilitating timely intervention when the prospects for successful resolution of kidney disease are greater. As highlighted in other phases of 0by25, most cases were related to infection, renal hypoperfusion, and nephrotoxins, and as such treatment occurred by relatively simple means [7,10]. Indeed, most patients were managed within the healthcare facility of patient presentation and need for transfer and KRT were low. This should be reassuring for healthcare systems considering implementation of the approach, and highlights that ensuring healthcare facilities have the necessary basic resource for treating infection and correction of altered volume status leads to effective treatment of most cases of AKD in LLMICs.

Follow-up rates across sites were impacted by the costs required for travel back to the healthcare facility, and as such exploring novel mechanisms for remote monitoring of kidney function is a research priority for LLMIC kidney care. Despite the challenges, follow-up rates (62% of the AKD patients across sites) were markedly improved compared to prior to project implementation. Unexpected was the relatively high post-discharge mortality rate in addition to the high proportion of surviving patients with persistent kidney disease, many of whom fulfilled a new diagnosis of CKD. It is evident that patient follow-up post AKD episode needs to be part of an overarching strategy for management of both acute and chronic kidney disease.

This paper is a narrative summary, and the project was designed to assess feasibility in protocol implementation within routine clinical care. As such, we largely present details of a service improvement initiative, and this project was not designed as a clinical study or trial. We have focused on a description of the feasibility of implementing the protocol, but this does not extend to a formal analysis incorporating implementation science, nor do we present an analysis of the use of the risk score, or patient outcomes in those included in the project. We did not collect data on facility length of stay. Not all patients presenting to healthcare facilities within the timeframe of the project were screened for kidney disease, given the practical challenge of large numbers of patients presenting to these settings, and there may have been selection bias in terms of who was included. Most patients were screened using the risk score, however clinical judgement was used in some centres and there was heterogeneity in this clinical approach between sites. We do not formally compare clinical practice pre and post project implementation, but the approach provides clear benefits in HCCs that had no prior access to kidney function testing. We don't have data to support a comparative analysis in the hospital settings included in this project, however previous published studies have demonstrated that only a minority of hospitalised patients have kidney function testing in LLMICs elsewhere [20]. Further anecdotal support of this comes from a previous audit of kidney disease management previously undertaken by us in unselected general medical admissions at a central hospital in Malawi; a decision to measure serum creatinine was made in 67.3% of patients, but this sample was only taken and processed in 53.3% and 21.3% of patients respectively (unpublished data). As such, we feel the strategy adds value even in the hospital setting in LLMICs, where timely assessment of kidney status is certainly not routine.

True success of project implementation will be determined by its sustainability beyond this initial period of data collection. This will require ongoing engagement of local communities in addition to advocacy with governments that determine healthcare priorities. Ultimately, we hope to be able to provide further economic analysis of the approach to demonstrate it to be a cost-effective mechanism for improving population health. In summary, we have provided a description of the successful implementation of a novel strategy for the management of AKD as part of routine healthcare in LLMICs. We advocate for the approach to become the new standard of care for AKD management in these settings.

### Inclusivity in global health research

Additional information regarding the ethical, cultural, and scientific considerations specific to inclusivity in global research is included in the Supporting Information (S1 File).

### Key messages

**What was known.** AKD is common in patients presenting to healthcare facilities in LLMICs. AKD outcomes are poor in these settings due to a lack of training of healthcare workers, limited availability of SCr measurement for AKD detection, and the absence of an accepted approach for its management thereafter.

**What this study adds.** A strategy to identify and manage AKD in low-resource settings was developed based on the use of a symptom-based risk score, point-of-care creatinine measurement, and education of healthcare workers. This novel strategy was feasible when implemented as part of routine healthcare in a large number of patients in multiple LLMICs worldwide.

**Potential impact.** The successful implementation of the approach highlights the feasibility of improving AKD management in LLMICs by relatively simple means. The management strategy has the potential to become the new standard of AKD care in LLMICs. Its future use will require advocacy efforts with stakeholders involved in providing resource for kidney care.

## Supporting information

**S1 Table. Study sites for KCN project implementation.**
(DOCX)

**S2 Table. Definitions of kidney disease.**
(DOCX)

**S1 File. Inclusivity in global health research.**
(DOCX)

## Acknowledgements

We thank the healthcare workers who provided clinical care during this study and, above all, the patients for their participation. We acknowledge the work of the ISN for their administrative support throughout.

## Author contributions

**Conceptualization:** Rhys D.R. Evans, Rolando Claure-Del Granado, Brett Cullis, Emmanuel A Burdmann, Bhupendra Shah, David C. Harris, Mike V. Rocco.

**Data curation:** Rhys D.R. Evans.

**Formal analysis:** Rhys D.R. Evans.

**Funding acquisition:** David C. Harris, Mike V. Rocco.

**Investigation:** Sanjib K Sharma, Rolando Claure-Del Granado, Brett Cullis, Emmanuel A. Burdmann, FOS Franca, Junio Aguiar, Martyn Fredlund, Mamit Rai, Shyam Kafle, Bhupendra Shah, Mike V. Rocco.

**Project administration:** Sanjib K. Sharma, Rolando Claure-Del Granado, Brett Cullis, Emmanuel A Burdmann, FOS Franca, Junio Aguiar, Martyn Fredlund, Kelly Hendricks, Maria F. Iturricha-Caceres, Mamit Rai, Mike V. Rocco.

**Resources:** Kelly Hendricks, David C. Harris.

**Writing – original draft:** Rhys D.R. Evans, Mike V. Rocco.

**Writing – review & editing:** Rhys D.R. Evans, Sanjib K. Sharma, Rolando Claure-Del Granado, Brett Cullis, Emmanuel A. Burdmann, FOS Franca, Junio Aguiar, Martyn Fredlund, Kelly Hendricks, Maria F. Iturricha-Caceres, Mamit Rai, Shyam Kafle, Bhupendra Shah, David C. Harris, Mike V. Rocco.

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
