## [Decision Letter · Decision Letter 0]

29 Oct 2024

PONE-D-24-36949Management of acute kidney disease as part of routine clinical care in low-resource settings: The International Society of Nephrology Kidney Care Network ProjectPLOS ONE

Dear Dr. Evans,

Thank you for submitting your manuscript to PLOS ONE. After careful consideration, we feel that it has merit but does not fully meet PLOS ONE’s publication criteria as it currently stands. Therefore, we invite you to submit a revised version of the manuscript that addresses the points raised during the review process.

We look forward to receiving your revised manuscript.

Kind regards,

Edward Zimbudzi

Academic Editor

PLOS ONE

Journal Requirements: When submitting your revision, we need you to address these additional requirements. 1. Please ensure that your manuscript meets PLOS ONE's style requirements, including those for file naming. The PLOS ONE style templates can be found at https://journals.plos.org/plosone/s/file?id=wjVg/PLOSOne_formatting_sample_main_body.pdf and https://journals.plos.org/plosone/s/file?id=ba62/PLOSOne_formatting_sample_title_authors_affiliations.pdf 2. Please include a complete copy of PLOS’ questionnaire on inclusivity in global research in your revised manuscript. Our policy for research in this area aims to improve transparency in the reporting of research performed outside of researchers’ own country or community. The policy applies to researchers who have travelled to a different country to conduct research, research with Indigenous populations or their lands, and research on cultural artefacts. The questionnaire can also be requested at the journal’s discretion for any other submissions, even if these conditions are not met.  Please find more information on the policy and a link to download a blank copy of the questionnaire here: https://journals.plos.org/plosone/s/best-practices-in-research-reporting. Please upload a completed version of your questionnaire as Supporting Information when you resubmit your manuscript. 3. Thank you for stating the following in the Competing Interests section: "I have read the journal's policy and the authors of this manuscript have the following competing interests: RE has received honoraria from Therakos. EB has received honoraria from Baxter and AstraZeneca." Please confirm that this does not alter your adherence to all PLOS ONE policies on sharing data and materials, by including the following statement: ""This does not alter our adherence to  PLOS ONE policies on sharing data and materials.” (as detailed online in our guide for authors http://journals.plos.org/plosone/s/competing-interests).  If there are restrictions on sharing of data and/or materials, please state these. Please note that we cannot proceed with consideration of your article until this information has been declared.  Please include your updated Competing Interests statement in your cover letter; we will change the online submission form on your behalf. 4. Your ethics statement should only appear in the Methods section of your manuscript. If your ethics statement is written in any section besides the Methods, please move it to the Methods section and delete it from any other section. Please ensure that your ethics statement is included in your manuscript, as the ethics statement entered into the online submission form will not be published alongside your manuscript. 5. Please include your tables as part of your main manuscript and remove the individual files. Please note that supplementary tables (should remain/ be uploaded) as separate ""supporting information"" files

Reviewers' comments:

Reviewer's Responses to Questions

**Comments to the Author**

1. Is the manuscript technically sound, and do the data support the conclusions?

Reviewer #1: Partly

Reviewer #2: No

2. Has the statistical analysis been performed appropriately and rigorously? 

Reviewer #1: No

Reviewer #2: No

3. Have the authors made all data underlying the findings in their manuscript fully available?

Reviewer #1: Yes

Reviewer #2: No

4. Is the manuscript presented in an intelligible fashion and written in standard English?

Reviewer #1: Yes

Reviewer #2: Yes

5. Review Comments to the Author

Reviewer #1: I thoroughly enjoyed reading this important study

Introduction is clear. Aim is clearly defined

The KCN approach is nicely summarized.

How did they determine difference between AKI and AKD is recommended to be verbalized in the text.

Methods - Some clear description of article type is recommended. Is this a narrative summary?

Some clarity on difference between HCCs, district hospitals, and tertiary hospital is needed. I see it mentioned in the text eventually. But highlighting how this approach can be used in HCC would be strength

What was the implementations strategy used? Perhaps a framework?

Results and discussion

The team states “successful implementation of the KCN approach” – What were the parameters used to describe successful implementation?

The way I read the paper, it is a very descriptive summary evaluating feasibility of this approach. I would downplay the implementation aspect as implementation science approaches necessitates a more systematic evaluation of “success” parameters, fidelity etc.

The summary from each country could benefit from a uniform and structured organization. scope, patients, etc.

I would highlight that this approach led to more female patients being screened.

Some formatting comments

Would minimize use of abbreviations. There are many that are being used and the word is being spelt out again. For example, Page 7 line 215 KRT is spelt out again

Reviewer #2: This is a report on the findings following the implementation of a programme i.e. the Kidney Care Network project under the International Society of Nephrology's 0by25 initiative on reducing to zero number of the deaths that occur globally in people who would have developed acute kidney injury(AKI) which is supposed to be reversible in most instances. Most of these deaths have been occurring in resource limited settings. The cause of the AKI is usually secondary to conditions that are easily amenable to treatment with e,g just rehydration and treatment of sepsis. The authors claim to have implemented the project successfully but there are no data, even historical data, that the authors have shared in this article, for them to compare their current findings to. Thus there are no The manuscript is just a description/narrative of the findings in "4 healthcare centres across 3 continents".

There is no definition of Acute Kidney Disease in this paper. The sampling strategy is not clear. Patients were screened for this renal dysfunction using a risk scoring system but we are also told that some were identified using a clinical assessment. Did the researchers include (screen) all patients presenting at these facilities where they were conducting this project? If not who was screened? Were those screened already "screened" clinical by the healthcare workers on the ground? Or were all consecutively seen patients screened? This explicit definition of this target population would allow the reader to be able to assess how these results can be "referred" back to the rest of the population.

The project was conducted in these 4 healthcare centres (HCC) in Bolivia, Brazil, Nepal and South Africa that were not similar. For instance in South Africa, the HCC was basically a primary HCC which one would expect not to to be able to have easy access to renal function tests and hence would have benefited from the POC testing for serum creatinine/educational activities being offered unlike the Nepal sites which was a higher level of care setting. We have no previous data to allow to judge if the findings of this project as presented were an improvement on the service delivery outcomes before this project came on board.

The authors refer to patients having been discharged but we are not told for instance for South Africa what this discharge meant. Were patients admitted and kept in the HCC for days as is likely to have been the case in perhaps Nepal a tertiary centre? These details need to be clarified.

The follow up of participants/ patients after 90 days does explain why the authors were using the term AKD as opposed to AKI.

This paper ideally should be reported using the following format: 1) Introduction 2) Project description 3) Project findings as opposed to using Discussion.

As currently presented, the manuscript is missing lots of detail in the project description which the author says is for another paper and not for this paper. The Discussion is just a narrative, The author refers to successful implementation of this project without summarizing the exact findings/results to support this assertion. What is the evidence for this statement? The educational activity was reported as having been conducted "regularly". What does this mean. There was no comparing or contrasting on this project findings to other findings in the wider literature.

6. PLOS authors have the option to publish the peer review history of their article (what does this mean? ). If published, this will include your full peer review and any attached files.

**Do you want your identity to be public for this peer review?** For information about this choice, including consent withdrawal, please see our Privacy Policy .

Reviewer #1: No

Reviewer #2: No

---

## [Author Response · Author response to Decision Letter 0]

16 Nov 2024

Please see uploaded response to reviewers document

---

## [Decision Letter · Decision Letter 1]

3 Dec 2024

Management of acute kidney disease as part of routine clinical care in low-resource settings: The International Society of Nephrology Kidney Care Network Project

PONE-D-24-36949R1

Dear Dr. Evans,

We’re pleased to inform you that your manuscript has been judged scientifically suitable for publication and will be formally accepted for publication once it meets all outstanding technical requirements.

Kind regards,

Edward Zimbudzi

Academic Editor

PLOS ONE

Additional Editor Comments (optional):

Reviewers' comments:

Reviewer's Responses to Questions

**Comments to the Author**

1. If the authors have adequately addressed your comments raised in a previous round of review and you feel that this manuscript is now acceptable for publication, you may indicate that here to bypass the “Comments to the Author” section, enter your conflict of interest statement in the “Confidential to Editor” section, and submit your "Accept" recommendation.

Reviewer #1: All comments have been addressed

Reviewer #2: All comments have been addressed

2. Is the manuscript technically sound, and do the data support the conclusions?

Reviewer #1: Yes

Reviewer #2: Yes

3. Has the statistical analysis been performed appropriately and rigorously? 

Reviewer #1: N/A

Reviewer #2: N/A

4. Have the authors made all data underlying the findings in their manuscript fully available?

Reviewer #1: Yes

Reviewer #2: No

5. Is the manuscript presented in an intelligible fashion and written in standard English?

Reviewer #1: Yes

Reviewer #2: Yes

6. Review Comments to the Author

Reviewer #1: All comments have been addressed. I am not sure what else to write here to meet the 100 character limit

Reviewer #2: 2-The authors state clearly in their response to the reviewers that this was not a clinical trial but a narrative summary of their findings on a project. The project/programme has been clearly described.

3- No statistical comparisons were conducted for this work.

4- The authors state that the data belongs to the International Society of Nephrology and would be made available given a reasonable request by researchers!

7. PLOS authors have the option to publish the peer review history of their article (what does this mean? ). If published, this will include your full peer review and any attached files.

**Do you want your identity to be public for this peer review?** For information about this choice, including consent withdrawal, please see our Privacy Policy .

Reviewer #1: No

Reviewer #2: No

---

## [Editor Report · Acceptance letter]

PONE-D-24-36949R1

PLOS ONE

Dear Dr. Evans,

I'm pleased to inform you that your manuscript has been deemed suitable for publication in PLOS ONE. Congratulations! Your manuscript is now being handed over to our production team.

Kind regards,

on behalf of

Dr. Edward Zimbudzi

Academic Editor

PLOS ONE